# Application of Metabolomics in Obesity-Related Childhood Asthma Subtyping: A Narrative Scoping Review

**DOI:** 10.3390/metabo13030328

**Published:** 2023-02-23

**Authors:** Heidi Makrinioti, Zhaozhong Zhu, Carlos A. Camargo, Valentina Fainardi, Kohei Hasegawa, Andrew Bush, Sejal Saglani

**Affiliations:** 1Department of Emergency Medicine, Massachusetts General Hospital, Harvard Medical School, Boston, MA 02114, USA; 2Clinica Pediatrica, Department of Medicine and Surgery, University of Parma, 43126 Parma, Italy; 3National Heart and Lung Institute, Imperial College, London SW7 2AZ, UK; 4Centre for Paediatrics and Child Health, Imperial College, London SW7 2AZ, UK; 5Royal Brompton Hospital, London SW3 6NP, UK

**Keywords:** obesity, asthma, subtyping, metabolomics, endotypes

## Abstract

Obesity-related asthma is a heterogeneous childhood asthma phenotype with rising prevalence. Observational studies identify early-life obesity or weight gain as risk factors for childhood asthma development. The reverse association is also described, children with asthma have a higher risk of being obese. Obese children with asthma have poor symptom control and an increased number of asthma attacks compared to non-obese children with asthma. Clinical trials have also identified that a proportion of obese children with asthma do not respond as well to usual treatment (e.g., inhaled corticosteroids). The heterogeneity of obesity-related asthma phenotypes may be attributable to different underlying pathogenetic mechanisms. Although few childhood obesity-related asthma endotypes have been described, our knowledge in this field is incomplete. An evolving analytical profiling technique, metabolomics, has the potential to link individuals’ genetic backgrounds and environmental exposures (e.g., diet) to disease endotypes. This will ultimately help define clinically relevant obesity-related childhood asthma subtypes that respond better to targeted treatment. However, there are challenges related to this approach. The current narrative scoping review summarizes the evidence for metabolomics contributing to asthma subtyping in obese children, highlights the challenges associated with the implementation of this approach, and identifies gaps in research.

## 1. Introduction

Obesity and asthma are complex diseases with increasing prevalence in children and an associated global public health impact [1,2]. Several prospective cohort studies and a systematic review and meta-analysis identify obesity as a risk factor for childhood asthma development [3,4,5]. In addition, obese children have worse asthma control, increased number of asthma exacerbations, and overall lower quality of life compared to children with a healthy weight [6]. 

It has been suggested that obesity-related childhood asthma is a distinct phenotype [7]. The variability in airway inflammatory pathways associated with this childhood asthma phenotype is likely related to the complexity of gene-environment interactions and has implications for therapy [8,9]. Therefore, by understanding gene–environment interactions, we may be able to define more homogeneous groups in terms of both clinical features and underlying pathophysiology (i.e., subtypes) in obese children with asthma [8,10]. A subtyping approach aims to facilitate precision medicine, thereby improving the response to treatment and prevention of chronic respiratory sequelae [11].

Environmental risk factors for obesity-related childhood asthma include physical activity, diet, indoor and outdoor allergens, air pollutants, tobacco, e-cigarette smoke, respiratory viruses and bacteria [12]. The mechanisms underlying interactions between these environmental factors and childhood asthma development are complex. However, with the recently increasing use of “omics” analytical methodologies, we can investigate disease mechanisms using less invasive biospecimens from children [13,14].

Metabolomics (i.e., the study of the metabolome of cells, tissues, organs, or organisms) is the most commonly applied analytical “omic” methodology for investigation of environment–host interactions [15,16]. Metabolites reflect variations in the time course of the metabolic activity of cells, tissues, organs, and organisms [17]. Recent evidence shows that more than 60% of blood metabolites are associated with either the host genome or gut microbiome [18]. Hence, metabolomics can be used to trace complex molecular pathways and quantify the impact of gene-environment interactions on childhood asthma development and control [19,20,21]. Several observational studies have identified distinct metabolic signatures more prevalent in obese children with asthma or reduced asthma control [22,23]. For example, increased insulin resistance and lower plasma levels of histidine and glutamine are more prevalent in obese children with asthma, whilst decreased serum long-chain n3 polyunsaturated fatty acid levels in obese children with asthma are associated with reduced asthma control [24,25]. However, metabolic signatures usually refer to a particular timepoint of sample collection and may fail to accurately represent complex interactions over time [26]. More evidence is needed to help us understand how to better apply metabolomics in a precision medicine approach in obesity-related childhood asthma.

This is a scoping review discussing past and current evidence on the application of metabolomics in subtyping obesity-related childhood asthma. Relevant studies have been identified through query of the Medline, Embase, and Cochrane databases for English language articles published until January 2023 using the terms “metabolomics”, “metabolites”, “metabolic dysregulation”, “asthma”, “wheezing”, “wheeze”, “obese”, “obesity”, “overweight”. Studies were selected for discussion in this review based on topical relevance. Publications cited by articles identified through the search strategy were included as appropriate. 

To our knowledge, this is the first review focusing on the application of metabolomics in childhood obesity-related asthma subtyping. A recently published systematic review has addressed a similar topic in childhood asthma but lacks the focus on obesity [21]. Our narrative scoping review follows a distinct structure by discussing relevant evidence around the epidemiology, respiratory physiology, and genetic susceptibility of obesity-related childhood asthma, themes not addressed in the systematic review cited above. In addition, following discussion around both inflammatory and metabolic mechanisms underlying obesity-related childhood asthma pathogenesis, our narrative scoping review conceptualizes a model of obesity-related childhood asthma subtyping possibly facilitated by metabolomics in the future. Our review concludes with suggestions around the design of future research studies that aim to bridge identified gaps.

## 2. Relevant Sections

### 2.1. Evidence around the Epidemiology, Respiratory Physiology and Genetic Susceptibility of Obesity-Related Childhood Asthma

#### 2.1.1. Emerging Epidemiological Data on Obesity-Related Childhood Asthma

The first epidemiological data associating obesity and asthma came from cohort studies in adults. Initial observations identified that adults with asthma have a higher risk of being obese (or developing obesity) for various reasons (e.g., reduced exercise, prescription of oral steroids) [27]. Camargo et al. were the first to show that body mass index (BMI) is prospectively associated with an increased risk for adult-onset asthma [28]. Since then, prospective cohort studies in children followed up until school-age or early adulthood have confirmed this association [29]. For example, a recent cohort study of infants hospitalized with severe bronchiolitis showed that children with a persistent obese profile have a higher risk of incident asthma by six years compared to children with a transiently obese or healthy weight profile [30]. In support of these observational data, a meta-analysis reported that childhood obesity is associated with a 50% increased odds of asthma development compared to lean children [7]. Several cohort studies suggest that this association starts in utero [31,32]. A meta-analysis revealed that maternal obesity and weight gain during pregnancy are associated with a 30% increased risk of asthma development in the offspring [33]. This study also suggested that metabolites secreted during pregnancy (e.g., serum leptin) may mediate asthma development in the offspring. In addition to these epidemiological data, there is increasing evidence showing other factors are involved in the causal pathway linking obesity to asthma development [34]. These factors (e.g., evidence of allergic sensitization, systemic inflammation) may act either as confounders or effect modifiers. 

Further epidemiological evidence highlights that children with obesity-related asthma have worse asthma control compared to non-obese children with asthma [35]. Moreover, there is a reduced response to conventional treatment (e.g., inhaled corticosteroids) in some obese children (more commonly boys) [36]. However, the direction of this association is unclear, as there is growing evidence that children with poor asthma control tend to exercise less and thus have a higher BMI compared to children with well-controlled asthma [37]. Research studies focusing on exploring respiratory physiology in obese children with asthma attempt to provide insight into the pathophysiological mechanisms underlying these associations.

#### 2.1.2. Respiratory Physiology Profiles in Obesity-Related Childhood Asthma

Research studies utilizing pulmonary function testing in obese children with asthma have described varying profiles of pulmonary function deficits. More specifically, analysis of spirometry findings shows that obese children with asthma have a lower ratio of Forced Expiratory Volume at 1 s to Forced Vital Capacity (FEV1/FVC) than non-obese children with asthma. This lower ratio can be explained by either lower FEV1 with normal FVC, or slightly lower FEV1 and FVC secondary to airway dysanapsis [38,39]. Airway dysanapsis is a physiological phenomenon likely attributed to the lungs being larger and the airways longer but of normal caliber [39]. Airway dysanapsis is more common in obese children with asthma and increased central fat and is associated with more frequent asthma attacks [39]. Pathophysiological mechanisms linking central adiposity (i.e., increased visceral fat) to non-type2 asthma, further supporting variation in respiratory physiology data in obese children with asthma, are discussed in the following section.

Interestingly, only a few studies have attempted to quantify lung volumes in obese children with asthma despite the anticipated impact of thoracic and abdominal fat deposition on lung volumes [40]. However, there are few reports describing reduced functional residual capacity and expiratory reserve volume in obese patients, including children, suggesting an impact of obesity on pulmonary function deficits starting early in life [41,42]. In summary, research studies describing respiratory physiology profiles in children highlight that there is variation in the pathophysiological mechanisms underlying these profiles that could possibly be better explained by understanding further genetic and other molecular pathways.

#### 2.1.3. Genetic Susceptibility in Obesity-Related Childhood Asthma

Some common features between obesity and asthma can be partially explained by an existing genetic association between the two conditions [43,44]. This genetic association seems to be stronger between childhood BMI and asthma development rather than adult BMI and asthma development, implying that early-life changes in BMI are an especially important risk factor for childhood asthma development [45]. In addition, a genome-wide association study (GWAS) from the Childhood Asthma Management Program (CAMP) found a shared genetic predisposition to obesity and asthma in children [43]. Additionally, a cross-trait GWAS in more than 100,000 subjects from the China Kadoorie Biobank identified shared genetic loci between obesity and impaired lung function [46]. These genetic loci are involved in biological pathways related to adipocyte proliferation, adipose tissue deposition, and regulation of glypican-5 (GPC5) gene expression. However, a shared genetic etiology between obesity and impaired lung function traits does not fully explain variability in the risk of developing asthma and asthma severity in obese children. More specifically, observational studies describe cases of obese siblings who differ in asthma status or severity [4,47]. Therefore, a better understanding of the inflammatory and metabolic pathways underlying obesity-related childhood asthma may provide supportive mechanistic evidence to the epidemiological observations. 

### 2.2. Inflammatory and Metabolic Mechanisms Underlying Obesity-Related Childhood Asthma Subtypes

#### 2.2.1. Inflammatory Mechanisms 

The commonest inflammatory endotype described in children with asthma includes the endotype characterized by type2-high immune profiles with predominant eosinophilia [48]. However, observational studies have described predominantly type 2-low immune profiles in obese children above 12 years old who have an asthma diagnosis [49,50]. Type 2-low immune profiles can be further characterized by airway non-eosinophilic (e.g., neutrophilic or basophilic) inflammation [51]. 

##### Role of Obesity in Inflammation in Children with Asthma

Obesity is a proinflammatory state and is associated with systemic low-grade inflammation characterized by increased proinflammatory cytokine secretion from adipose tissue and infiltration of leukocytes, including macrophages, into adipose tissue [52]. Adiposity is the main driver of the proinflammatory response and is associated with chronic inflammation in obese children [53,54]. Adipose tissue is a connective tissue that extends under the skin (i.e., subcutaneous), between internal organs (i.e., visceral), and in the inner cavities of bones (i.e., bone marrow). Visceral adipose tissue is an important source of cytokines [55]. Case-control studies have highlighted that inflammatory mediators, such as C-reactive protein (CRP), tumor necrosis factor-alpha (TNF-α), and interleukin-6 (IL-6) concentrations are increased in serum from children with increased visceral fat [56]. In addition, visceral fat is an important source of adiponectin, while subcutaneous fat is an important source of leptin [57]. Adiponectin and leptin have been considered to have different associations with asthma endotypes [58]. Leptin, an adipokine, is secreted in high levels from expanded adipose tissue and has proinflammatory properties, whilst adiponectin is secreted in lower levels in response to expanded adipose tissue and has anti-inflammatory properties [59]. In general, the pathological proliferation of adipocytes is a source of inflammation within the adipose tissue [60]. Mechanistic studies suggest that adipocyte-triggered chronic inflammation may also underlie childhood asthma development [61,62]. However, in contrast to myeloid cells, regulation of adipocyte proinflammatory mechanisms is poorly understood. Therefore, part of the variability in asthma endotypes in obese children can be attributed to body fat distribution [63]. The body fat distribution and the use of different fat measures may also explain heterogeneous evidence around systemic and airway inflammation in obese children with asthma [64]. This is demonstrated in a Canadian study of asthma in 54 children [65]. Interestingly, all four anthropometric parameters (BMI, waist-to-hip ratio, neck and midarm circumference) assessed in 54 children with asthma had distinct and interdependent associations with incident asthma [65].

##### Interferons Contributing to Airway Inflammation in Obese Children with Asthma

There is some evidence from mechanistic studies that might provide further clarity about the proinflammatory potential of adipocytes. These studies show that adipocytes secrete proinflammatory cytokines, express innate immune receptors (e.g., Toll-like receptor (TLR)-7) and are antigen-presenting cells. Type I interferons (IFNs), predominantly IFNα and IFNβ, are immune mediators that orchestrate both innate and adaptive immune responses [66]. IFNβ is expressed by both myeloid cells and adipocytes, while myeloid cells are the primary source of IFNα [67]. Obesity-associated metabolic endotoxemia (lipopolysaccharide-induced) activates TLR signaling, that in turn induces IFN type I secretion [68]. IFNα transmembrane receptor (IFNAR) initiates and mediates multiple inflammatory signaling pathways. The literature on the role of obesity-associated IFN signaling is contradictory. Some studies have identified a protective role for IFN signaling in obesity-related asthma [69,70]. More specifically, some studies propose that upregulated IFN-related signaling in obese children is associated with reduced risk for incident asthma [71]. In contrast, a recent mechanistic study showed the IFNAR axis triggers glycolysis-associated adipocyte inflammatory signaling similar to that of myeloid cells [72]. These observations highlight the potential role of the IFNAR axis in systemic inflammation in obese children. Figure 1 describes a model of the role of type I IFN/IFNAR axis in mediating adipocyte-driven inflammatory processes underlying persistent airway inflammation in obese children.

##### Role of Obesity in Airway Remodeling in Children with Asthma

Mechanistic studies have shown that obesity induces airway epithelium remodeling via either excessive deposition of adipose tissue in the outer epithelial walls of medium and large airways or via leptin-mediated mechanisms. More specifically, leptin triggers the expression of the intercellular adhesion molecule (ICAM-1) in bronchial epithelial cells and the proliferation and migration of airway epithelial cells. The different mechanisms that underlie airway remodeling in obese in vivo models of asthma may explain the endotypic heterogeneity [73]. Moreover, the adipose tissue from obese children is a source of proinflammatory mediators that may underlie chronic airway inflammation and airway remodeling [54,61,74].

However, the associations between body fat-driven systemic and airway inflammation and remodeling can explain only some of the identified asthma endotypes in obese children. Metabolic dysregulation of various types (e.g., insulin resistance and/or dyslipidemia) provides further insight into the endotypic characterization of obesity-related asthma by identifying specific profiles (i.e., subtypes) of asthma in obese children.

#### 2.2.2. Metabolic Mechanisms

Oxidative stress is a central pathogenetic process in obesity-related childhood asthma and is considered to drive other metabolic processes, including dysregulation of fatty acids, peripheral blood ketones, amino acid metabolism, and insulin resistance [24,25,41,65,75].

Studies on ex vivo models of obstructive airway disease have identified that oxidative stress, through its impact on mitochondrial function, drives airway inflammation and remodeling [76,77,78]. More specifically, in response to ozone stimulation, mitochondrial dysfunction triggers the development of airway inflammation and hyperresponsiveness [79]. In addition to ozone, other free radicals (i.e., reactive oxygen species and reactive nitrogen species) are considered significant mediators of airway tissue damage and are more prevalent in subjects with asthma [80,81]. For instance, glutathione secretion is considered to exert a protective role for the airway epithelium in response to oxidative stress. More specifically, a pattern of increased exhaled breath condensate levels of malondialdehyde (MDA) and reduced glutathione levels is more prevalent in children with asthma [82]. In addition to evidence deriving from studies that utilize airway samples, there are studies utilizing other samples (e.g., serum or urine) that have linked the pattern of increased MDA and decreased glutathione levels to increased asthma severity [83]. However, it is still unclear whether obesity amplifies these findings in children with asthma.

##### Oxidative Stress and Dysregulation of Free Fatty Acid Metabolism Pathways in Childhood Asthma

The pathway associating IFNAR signaling and free fatty acid metabolism in obese children may be important. The role of free fatty acids has been reported in several metabolomic studies in childhood asthma [23,25]. However, there is still controversy in regard to the associations with long-term outcomes (i.e., asthma) [10,84]. Till now, the evidence shows that decreased peripheral blood n-3 long-chain fatty acid levels are associated with an increased risk for obesity-related asthma [85]. However, sparse evidence shows that interventions with n-6 long-chain fatty acids may be associated with increased airway inflammation [86,87]. These ostensibly controversial findings can be possibly explained by the timing, sensitivity of measurement tools, or possible interactions with other measured metabolites but need to be further elucidated. 

In contrast to the evidence around the role of polyunsaturated long-chain fatty acids, data from observational studies and experimental models investigating the role of short-chain fatty acids point consistently toward a protective role in childhood asthma [88]. However, evidence around their role in obesity-related childhood asthma is missing. Short-chain fatty acids are synthesized by gut microbiota through the fermentation of undigested carbohydrates and dietary fibers in the gastrointestinal tract [89]. Observational studies show that decreased levels of short-chain fatty acids are associated with increased asthma severity [10]. Additionally, reduced secretion of short-chain fatty acids (e.g., acetate, propionate, and butyrate) secondary to gut microbiome changes is associated with an increased risk of childhood asthma development [90]. Although the protective role of short-chain fatty acid supplementation is yet to be confirmed, mouse models of short-chain fatty acid supplementation show a protective effect on asthma development in the offspring [91]. These findings are also supported by observational data associating a reduced fecal level of short-chain fatty acids during pregnancy with an increased risk of asthma development. Recent in vivo pre-clinical data also indicate protection against rhinovirus-induced respiratory tract infections (i.e., the most common trigger of asthma exacerbations) following intranasal administration of short-chain fatty acids. However, this study does not focus on obesity-driven airway inflammation [92].

##### Oxidative Stress and Peripheral Blood Ketones and Glucose Dysregulation in Obesity-Related Asthma

In addition to fatty acid metabolism pathways, there are observational studies that associate blood ketone levels, amino acid metabolism, and control of hepatic glucose release with obesity-related asthma development in children and adults [93,94]. More specifically, past observational studies have shown that increased energy demands and hypoxia (i.e., in asthma) are associated with an increase in the consumption of ketone bodies [25]. There are two main ketone bodies, acetoacetate, and beta-hydroxybutyrate, being detected at lower levels in obese children with asthma compared to controls. 

In regard to amino acid metabolism, lower levels of plasma histidine and glutamine have been associated with increased asthma incidence [25]. Histidine is an essential amino acid [95]. Lower histidine levels have also been detected in obese women without asthma [96]. In female animal models, supplementation with histidine has been shown to impact weight increase and improve airway inflammation and oxidative inflammatory responses driven by adipose tissue [97]. 

In addition, a higher prevalence of insulin resistance and consequent hyperglycemia has been reported in obese asthmatic children [93]. Past studies have shown that impaired insulin secretion during the first years of life has been related to an increased risk of asthma development [98]. Several studies report that insulin resistance, in combination with other factors (e.g., abdominal fat and metabolic syndrome) [93], is associated with reduced lung function in asthmatic and non-asthmatic children. Interestingly, these studies in the general adult population show that insulin resistance is associated with an obstructive rather than restrictive (i.e., obesity-related) decline in lung function. 

The pathophysiologic mechanisms underlying these findings are poorly understood. Mechanistic studies have shown that airway smooth muscle (ASM) cells express insulin receptors and develop a pro-contractile phenotype when exposed to high levels of insulin [99]. Based on these findings, we speculate that the reported associations between insulin resistance and asthma may be enhanced by obesity and that glucose regulation pathways may help subtyping obesity-related asthma in childhood. 

In conclusion, metabolic pathways may explain a large part of obesity-related asthma pathophysiology in children [23,25,41,42,65,100]. Although levels of other metabolites, including urinary prostaglandin D2 and urinary leukotriene E4, have been associated with increased prevalence of childhood asthma, there is no evidence related to associations with obesity-related asthma prevalence [101]. The metabolomics studies that have reported associations between childhood and adult asthma and obesity are depicted in Table 1, and the main metabolic profiles are shown in Figure 2a. Both fatty acid metabolism and glucose regulation pathways merit further investigation, ideally in integration with other “omic” data so that asthma endotypes can be better described. 

## 3. Discussion around Challenges in the Application of Metabolomics in Precision Medicine in Obesity-Related Childhood Asthma

Translational opportunities for metabolomics are critical for a successful application of precision medicine in obesity-related childhood asthma. Observational studies have identified metabolic pathways that are more or less prevalent in obese children with asthma. However, there are many challenges in translating these findings into biomarkers or therapeutic targets in obesity-related childhood asthma.

The first challenge lies in using targeted assays that measure metabolites in clinical practice. No single measurement platform can measure all metabolites simultaneously. In addition to the different measuring platforms, experts in the analysis of acquired data (i.e., analytical chemists, statisticians, data scientists, and bioinformaticians) are required. Even so, finding experts in analyzing data acquired through different platforms is challenging. For example, researchers trained in liquid chromatography—mass spectrometry often need the help of experts in bioinformatics for the optimal experimental design for individual metabolomics studies and the appropriate statistics to be employed. With an optimistic tone, there are novel measurement platforms that can possibly address the above challenges in the future. Semiconductor metal oxide-based chemo-resistant sensors are being used for a variety of different functions (e.g., monitoring of the type and concentration of products of gas combustion) due to their small size, portability, and cost-effectiveness [107]. These tools detect mediators (e.g., volatile organic compounds), specifically using exhaled breath. Agilent Seahorse is another novel measurement tool assessing the kinetic activity (i.e., rates) of two main ATP-producing pathways in live cells: Mitochondrial respiration and glycolysis. Consequently, the tool offers a dynamic rather than static measurement of metabolism and is currently used in live cell cultures and in tissue (i.e., bronchial tissue) samples [108].

The second challenge Involves the measurement of metabolites related to interactions between specific environmental exposures (e.g., exposure to high levels of traffic-related air pollution or passive smoking or unhealthy diet), and host genetic and epigenetic traits. Disentangling the numerous molecular pathways associated with each metabolic signature is complex and may require integration with other “omic” data (e.g., genomic or epigenomic) for a more holistic understanding. 

The third challenge is associated with biospecimen type, collection, and preservation. Detecting and measuring polar molecules and lipids with increased sensitivity, specificity. and reproducibility is difficult. Factors that can impact detection (e.g., timing of biospecimen collection, biospecimen collection procedure, processing, stabilization, and storage) are also associated with sampling techniques. We might achieve consistent and more sensitive measurements (i.e., better detection yield) and easy validation of metabolomics-related biomarkers by utilizing novel sampling techniques (e.g., nasal scrapes, nasosorption). In summary, identifying metabolites as biomarkers for obesity-related childhood asthma remains an analytical challenge.

Regarding the identification of novel therapeutic targets based on metabolomics methodologies, observational and mechanistic studies in obesity-related childhood asthma show that fatty acid metabolism and glucose regulation pathways could be targeted for novel therapeutics in obesity-related childhood asthma. Free fatty acid receptors (FFARs) are a recently discovered class of G protein-coupled receptors (GPCRs) that are associated with several tissue-specific responses to dietary fatty acids (94). The interrelationship between FFAR and IFN signaling provides a possible insight into asthma pathophysiology. FFAR treatment induces endoplasmic reticulum (ER) stress response and downregulates the IFNAR1 chain of the type I IFN receptor leading to defective Jak-Stat signaling and impaired antiviral response [109]. Recent findings suggest that FFAR1 is expressed in airway smooth muscle cells and plays a pivotal role in airway contraction and airway smooth muscle cell proliferation. FFAR4 is also expressed in airway smooth muscle cells but does not contribute to airway contraction and airway smooth muscle cell proliferation. Both fatty acid metabolism and IFN regulation are impaired in children with asthma [23,110]. 

Therefore, the FFAR-IFN-related pathway may provide novel therapeutic targets for asthma in obese children. Glucose regulation pathways are metabolic pathways implicated in obesity-related asthma pathogenesis too. Metformin, drug targeting glucose regulation pathways, has been reported to affect allergic airway inflammation in in vivo models of obesity-related asthma [111]. This is another novel therapeutic target to be explored in obese non-diabetic children with asthma. 

## 4. Conclusions

In conclusion, metabolites represent the fingerprint of gene-environment interactions and can potentially capture phenotypic and endotypic differences in obesity-related childhood asthma. Metabolic pathway analysis could complement other targeted “omics” analyses to unravel mechanisms underlying inflammatory processes in obesity-related childhood asthma. However, interpreting absolute and total intracellular metabolite concentrations can be challenging, and future research should focus on developing sensitive metabolites measurement platforms. At all events, we should emphasize that the use of metabolomics and multiomics strategies should always be complementary to critical public health interventions (e.g., behavioral interventions toward weight management) in obese children with asthma. 

## 5. Future Directions of Research

A better understanding of the underlying complex pathophysiology in obesity-related childhood asthma could be achieved by systems biology approaches that integrate multiple types of quantitative molecular measurements (i.e., multiomics approaches). The current multiomics studies in obesity-related childhood asthma provide useful mechanistic insights. However, for a causal interpretation of these findings, we need cellular models studying the functional role of identified metabolic pathways. In addition to the design of multiomics studies, it is important to map metabolite excretion patterns in different specimens (peripheral blood, urine, feces) and develop metabolite measurement tools that can provide accurate specimen-specific measurements, if possible. In this way, applying metabolomics in obesity-related childhood asthma may pave the way for clinically relevant subtyping (Figure 2b) that will facilitate the development and testing of novel effective targeted treatments or drug repurposing. With respect to drug repurposing, a final suggestion from this review is to utilize mechanistic insights acquired through metabolomics studies and design clinical trials that will test approved drugs targeting more than one biological pathway (e.g., metformin acts on the insulin-stimulated pathway and a protein kinase-activated pathway affecting airway inflammation and remodeling). In summary, current research gaps include the design and implementation of multiomics studies that will provide novel mechanistic insights into obesity-related asthma pathogenesis, the development and testing of innovative metabolite measurement tools, and the design of clinical trials that utilize interventions based on mechanisms identified through multiomics studies.

## Figures and Tables

**Figure 1 metabolites-13-00328-f001:**
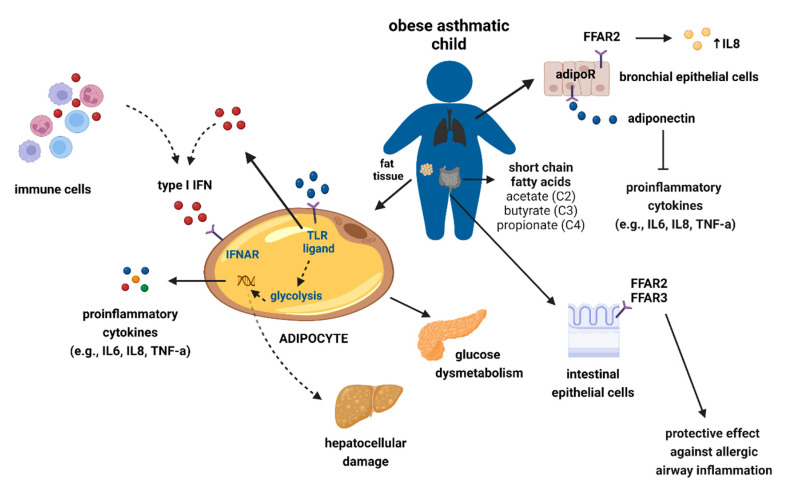
A representation of a model of the role of type I IFN/IFNAR axis in mediating adipocyte inflammatory processes underlying persistent airway inflammation in obese children. In fat tissue, adipocyte-specific IFNAR expression modifies the severity of obesity-associated glucose dysmetabolism with an impact on hepatocellular damage. More specifically, persistent type I IFN sensing may trigger T cells to become dysfunctional. The latter may explain why T cells show reduced proliferation and cytokines (e.g., IFN) release. In regard to inflammatory pathway mediation, adiponectin binds to adiponectin receptor AdipoR1 and reduces the activation of inflammatory pathways. FFAR2, a free fatty acid receptor, is expressed in bronchial epithelial cells and triggers the secretion of proinflammatory cytokines, more specifically IL8. Excessive production of proinflammatory cytokines and reduced secretion of IFNs may underlie the development of chronic airway inflammatory responses and explain observed associations between reduced short fatty acid secretion and asthma development in children. Abbreviations: FFAR2: Free fatty acid receptor 2, IFN: Interferon, IFNAR: IFN-alpha receptor, TLR: Toll-like receptor.

**Figure 2 metabolites-13-00328-f002:**
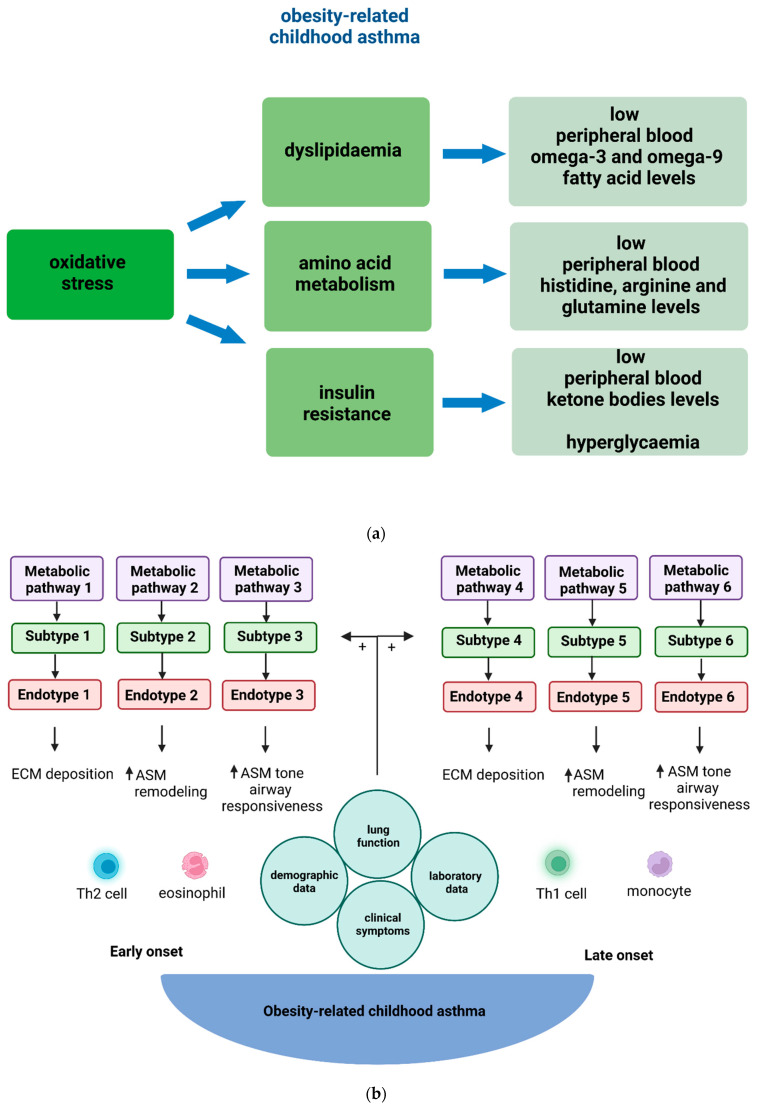
(**a**) Main metabolic profiles described as more prevalent in obese children with asthma in comparison to the control group (i.e., non-obese children with asthma). (**b**) The obesity-related childhood asthma phenotype consists of various endotypes. In children, early onset of obesity-related asthma is usually associated with T_H_2-high inflammation, whilst late onset of obesity-related asthma is associated with T_H_2-low inflammation. Metabolic dysregulation contributes to both T_H_2-high and T_H_2-low systemic and airway inflammation that underlies ECM deposition, ASM proliferation, related airway tissue remodeling, and airway reactivity in obesity-related asthma endotypes in children. Metabolomic pathways, identified by metabolomic strategies, provide further insight into the pathogenetic mechanisms underlying these diverse endotypes. Subtypes of obesity-related childhood asthma derive through the combination of clinical information with endotypic characterization data. Obesity-related asthma subtypes refer to groups of children with common clinical characteristics and endo-metabotypes (meaning a subtype of an endotype described by similar metabolic dysregulation mechanisms). Abbreviations: ASM: Airway smooth muscle, ECM: Extracellular matrix, T_H_2: T helper 2.

**Table 1 metabolites-13-00328-t001:** Table describing metabolomics studies that have reported associations between childhood and adult asthma and obesity.

Citation (Country)	Study Type	Study Population and Size	Exposure(s) or Intervention(s)	Metabolomics Technique	Cellular Location of Metabolic Pathway	Outcome (Incident Asthma or Prevalent Asthma or Asthma Control)	Measure of Association
studies in children							
Qu H.Q. et al. [25] United States	case-control study	603 children with asthma (stratification through weight level) and 593 children without asthma	peripheral blood samples ketone bodies, histidine, glutamine, saturated fatty acids, very low-density lipoprotein	NMR spectroscopy	endoplasmic reticulum, nucleus, cytoplasm, mitochondria	prevalent asthma by 18 years	lower plasma levels of histidine and glutamine more prevalent in asthma cases than in controls (*p* < 0.05) low peripheral blood ratio of ketone bodies, citrate, and fatty acids more prevalent in asthma cases with increased weight than in controls (*p* < 0.01)
Rastogi G. et al. [41] United States	case-control study	82 obese adolescents and 86 non-obese adolescents	peripheral blood samples insulin, lipids, adipokines	NMR spectroscopy	endoplasmic reticulum, cytoplasm, mitochondria	prevalent asthma by 16 years	increased adipokines and lipids levels are independently associated with reduced lung function and asthma prevalence by 16 years
Thompson D. et al. [65] United States	cohort study analytical study	26 children with obese asthma phenotype and 28 children with non-obese asthma phenotype	serum glucose, insulin, lipids, fatty acid levels and T_H_ cell transcriptome neck, WHR, and BMI z score	NMR spectroscopy	endoplasmic reticulum, nucleus, and mitochondria	asthma control	decreased lipids and increased fatty acid levels associated with increased asthma control and improved pulmonary function in obese children than in non-obese controls (*p* < 0.05)
Fitzpatrick A.M. et al. [83] United States	case-control study	257 children with asthma	peripheral blood samples leptin, adiponectin, C-reactive protein, total cholesterol, IL-1β, IL-6, IL-17, interferon gamma, tumor necrosis factor alpha, monocyte-chemoattractant protein-1, and amino acid metabolites	NMR spectroscopy	endoplasmic reticulum, cytoplasm, mitochondria	asthma control (6–17 years old)	within the group of obese children, lower concentrations of arginine-related metabolites associated with reduced asthma control lower vs higher (*p* <0.05)
Papamichael M.M et al. [24] Greece	case-control study	64 children with asthma	peripheral blood samples plasma fatty acid metabolites (linoleic, oleic, erucic, cis-11-eicosenoic, arachidic acids, α-linolenic, EPA and DHA)	GM-CS	endoplasmic reticulum, nucleus, cytoplasm, mitochondria	asthma control (5–12 years old)	peripheral blood α-linolenic, EPA and DHA levels not associated with reduced asthma control decreased level of linoleic, oleic, erucic, cis-11-eicosenoic, arachidic acids associated with reduced asthma control increased vs non-increased (*p* <0.05)
Tobias T.A.M. et al. [102] United States	case-control study	39 obese children with asthma, 39 normal weight children with asthma, 38 obese controls and 42 normal weight controls	peripheral blood samples plasma polyunsaturated fatty acids, carotenoids	NMR	endoplasmic reticulum, nucleus, and mitochondria	asthma control (13–18 years old)	increased level of peripheral blood polyunsaturated long-chain fatty acids correlated with improved asthma control (*p* < 0.01)
studies in adults							
Liu Y. et al. [103] China	case-control study	11 obese adults with asthma and 22 non-obese adults with asthma	peripheral blood and sputum samples peripheral blood cyanoaminoacid, caffeine, valine, uric acid, N-methy-DL-alanine and beta-glycerophosphoric acid metabolism sputum tryptophan and pentose phosphate metabolism	GC-MS	endoplasmic reticulum, nucleus, and mitochondria	prevalent asthma by 57 years (18–57 years old)	decrease in 3-hydroxybutyric acid, linolenic acid, isoleucine in obese vs non-obese adults with asthma (*p* < 0.05)
Rastogi D. et al. [100] United States	case-control study	334 overweight adults, and 648 obese adults	peripheral blood and sputum samples insulin resistance, HDL levels	Elisa in peripheral blood samples	endoplasmic reticulum, nucleus, and mitochondria	prevalent asthma by 60 years	high peripheral blood HDL levels associated with prevalent asthma in obese rather than overweight adults (*p* < 0.05)
Maniscalco M. et al. [104] Italy	case-control study	25 obese adults with asthma and 30 non-obese adults with asthma	EBC samples methane, glyoxylate/dicarboxylate, and pyruvate	NMR spectroscopy	cytoplasm, nucleus, and mitochondria	prevalent asthma by 50 years (30–50 years old)	EBC samples from obese patients with asthma had increased glucose, butyrate, and acetoin levels and decreased formate, tyrosine, ethanol, ethylene glycol, methanol, n-valerate, acetate, saturated fatty acids, and propionate levels as compared to non-obese patients with asthma (*p* < 0.004)
S. Y. Liao et al. [105] United States	randomized controlled trial	19 patients with severe asthma	peripheral blood samples intervention treatment with L-arginine or placebo at 0.05 mg/kg for 12 weeks, then 6-week washout period, and then treatment with L-arginine or placebo at 0.05 mg/kg exposures arginine-related metabolites, GLP-1, insulin	MS	endoplasmic reticulum, cytoplasm	asthma control (5–12 years old)	L-arginine supplementation was associated with increased insulin levels and decreased GLP-1 levels between those who received this and the control group (*p* = 0.02)
Mani M.L. et al. [106] United States	case-control study	19 healthy adults and 34 adults with asthma	peripheral blood samples bile acid levels (glycocholic acid and glycoursodeoxycholic acid)	LC-MS	endoplasmic reticulum, cytoplasm, and mitochondria	asthma control (18–65 years old)	increased peripheral blood glycocholic and glycoursodeoxycholic acid levels associated with reduced asthma control (*p* < 0.05)

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
