# Peer review of "Application of Metabolomics in Obesity-Related Childhood Asthma Subtyping: A Narrative Scoping Review"

_metabolites, 2023, doi:10.3390/metabo13030328_

Round 1

Reviewer 1 Report

This manuscript is a good review. I have some minor comments.

Lines 17, 34 and 116.

The word “exacerbation” needs to be mentioned at least once in this article, since recent guidelines use this term rather than “attack.”

Line 110.

The word “minute” should be changed to “second.”

Lines 117, 150, and others.

The phrase type 2 need to be used at least once in this manuscript, since the phrase TH2 asthma is not used commonly at present.

Line 280.

Probably a new Table 2, and not a textbox (Table 1) is necessary.

Line 369.

Jak-Stat should be changed to JAK-STAT.

Line 418.

TGF-b does not need to be mentioned in the figure legend, since it is not used in the figure.

Author Response

Dear Reviewer

Thank you kindly for taking the time to provide useful comments regarding our submitted manuscript titled: “Application of metabolomics in obesity-related childhood asthma subtyping: a narrative scoping review”.

We have addressed all the points you kindly made. Please see the answers below.

Comment 1

Lines 17, 34 and 116.

The word “exacerbation” needs to be mentioned at least once in this article, since recent guidelines use this term rather than “attack.”

Response to comment 1

Thank you for this suggestion. We have amended the text to include now the word “exacerbation” instead of the word “attack”.

Comment 2

Line 110.

The word “minute” should be changed to “second.”

Response to comment 2

Thank you for pointing this out. This change has been implemented.

Comment 3

Lines 117, 150, and others.

The phrase type 2 need to be used at least once in this manuscript, since the phrase TH2 asthma is not used commonly at present

Response to comment 3

Thank you for this suggestion. This change has been implemented in lines 125, 157, 158 and 159.

Comment 4

Line 280.

Probably a new Table 2, and not a textbox (Table 1) is necessary.

Response to comment 4

Thank you for this suggestion. We have removed the textbox and we have inserted the following text (lines 74-82):

“Relevant studies have been identified through query of the Medline, Embase and Cochrane databases for English language articles published until January 2023 using the terms “metabolomics”, “metabolites”, “metabolic dysregulation”, “asthma”, “wheezing”, “wheeze”, “obese”, “obesity”, “overweight”. Studies were selected for discussion in this review based on topical relevance. Publications cited by articles identified through the search strategy were included, as appropriate. To our knowledge, this is the first review focusing on the application metabolomics in childhood obesity-related asthma subtyping. As such, the review concludes with suggestions around design of future research studies that aim to bridge identified gaps”.

In addition, we have drawn a new figure, Figure 2a to summarize the main metabolomic profiles described in obese children with asthma.

Comment 5

Line 369.

     Jak-Stat should be changed to JAK-STAT.

Response to comment 5

Thank you for pointing this out. This change has been implemented.

     Comment 6

      Line 418.

TGF-b does not need to be mentioned in the figure legend, since it is not used in the  figure.

Response to comment 6

We apologize for this mistake. We have removed this word now.

Reviewer 2 Report

Article “Application of metabolomics in obesity-related childhood 2 asthma subtyping: a narrative scoping review” by Makrinioti et al. is about metabolomics which can help with defining clinically relevant obesity-related childhood asthma subtypes. In my opinion, the article still needs a lot of work.

COMMENTS:

at the end of the abstract there should be 1-2 sentences of conclusions

the citation in the text and final references should be adapted to the requirements of the journal

Authors should better describe how articles for review were selected - there is no information on how many articles were found, how many were rejected, from which years articles were searched.

the layout of the article is not typical of a review

overall, the authors focus little on metabolomics - there are too many references to topics unrelated to the main topic

a similar and better article was published recently https://doi.org/10.3390/metabo11040251

Author Response

Dear Reviewer

Thank you kindly for taking the time to provide useful comments regarding our submitted manuscript titled: “Application of metabolomics in obesity-related childhood asthma subtyping: a narrative scoping review”.

We have addressed all the points you kindly made. Please see the answers below.

Comment 1

at the end of the abstract there should be 1-2 sentences of conclusions

Answer to comment 1

Thank you for suggesting this.

The abstract has been edited to reflect this change (manuscript _ track changes on) in lines 25 to 32 (page 2). Kindly find new text attached below:

On a positive note, with the use of metabolomics, pathogenetic mechanisms underlying obesity-related asthma development will be further elucidated.  This will ultimately help define clinically relevant obesity-related childhood asthma subtypes that respond better to targeted treatment. However, there are challenges related to this approach. The current narrative scoping review summarises the evidence for metabolomics contributing to asthma subtyping in obese children, highlighting the challenges associated with the implementation of this approach and identifies gaps in research.”

Comment 2

the citation in the text and final references should be adapted to the requirements of the journal

Answer to comment 2

Thank you for pointing this out. We have adjusted the references format to the journal’s format (i.e., reference numbers in brackets, authors’ names, title of the article, abbreviated journal name year, volume, and page range).

Comment 3

Authors should better describe how articles for review were selected - there is no information on how many articles were found, how many were rejected, from which years articles were searched.

Answer to comment 3

Thank you for this suggestion. However, we would like to clarify that this is not a systematic review article. This is why we did not add a “Methods” section. We followed the recommendation for authors written under “article types” at the online site of the journal.

Following your suggestion though, we have added text describing the literature search at page 4, lines 73 to 82. Kindly find this text copied below:

Relevant studies have been identified through query of the Medline, Embase and Cochrane databases for English language articles published until January 2023 using the terms “metabolomics”, “metabolites”, “metabolic dysregulation”, “asthma”, “wheezing”, “wheeze”, “obese”, “obesity”, “overweight”. Studies were selected for discussion in this review based on topical relevance. Publications cited by articles identified through the search strategy were included, as appropriate.”

Comment 4

the layout of the article is not typical of a review

Answer to comment 4

We appreciate your comment. The suggested structure as per the journal includes the following sections; “Introduction, Relevant Sections, Discussion, Conclusions, and Future Directions”. This is the structure we have followed for the edited manuscript, and we have focused our discussion on challenges in the application of metabolomics in precision medicine in obesity-related childhood asthma, the main focus of the paper.

Comment 5

overall, the authors focus little on metabolomics - there are too many references to topics unrelated to the main topic

Answer to comment 5

Thank you very much for sharing your view.

We would like to emphasize that, to our knowledge, this is the first review focusing on the application of metabolomics on subtyping of obesity-related childhood asthma. This review is building on the review published by Rastogi D. et al. in 2016 (2016 May;137(5):e20150812. doi: 10.1542/peds.2015-0812. Epub 2016 Apr 8) with clear descriptions of metabotypes that are prevalent in obesity-related childhood asthma.

In the current form, there are 74 out of 101 references focusing on metabolomics (i.e., the study of chemical processes involving metabolites). We also discuss around epidemiological evidence (as a background) describing which questions have been answered in regard to obesity-related childhood asthma. These studies have been cited too and do not include metabolomics data.

Please see below our amended text at page 3 and lines 79-83:

“To our knowledge, this is the first review focusing on the application metabolomics in childhood obesity-related asthma subtyping. As such, the review concludes with suggestions around design of future research studies that aim to bridge identified gaps”.

Comment 6

a similar and better article was published recently https://doi.org/10.3390/metabo11040251

Answer to comment 6

We agree with you that this is a great review. We cited this. However, we need to emphasize that our review has a different focus – i.e., obesity-related childhood asthma. Furthermore, this excellent review was published in 2021, so there is scope for an update.

Reviewer 3 Report

The review by Makrinioti et al. entitled “Application of metabolomics in obesity-related childhood 2 asthma subtyping: a narrative scoping review” aims to guide the reader in understanding how metabolomics can help identifying endotypes of obese asthma. The review is worth publishing, after some modifications.

The unfortunate first impression from this review is that, in some sections, it lacks specificity, accent. There are too many general words and sometimes little specific details. Probably, like the consequence of many omics studies that are full of comparisons between groups but lack focused conclusion, the review lacks dedicated description of a few metabolic pathways dysregulated specifically in obese asthma but tries to discuss all possible information. However, the topic is difficult, and all said is only a mere “wish” but not a “request to change”.

There are a few specific points that need to be addressed.

page 7 line 280. Table 1 (that should be Table 2) is missing.

Section C (page 7, line 284) – either to move somewhere at the beginning of the manuscript or eliminate as it brings little information to understand how metabolomics can help to identify asthma endotypes.

p.8 line 351. The use of the phrase “metabolites are sensitive molecules” is misleading. There is oxidation, enzymatic cleavage, hydrolysis, as a few examples of processes that can be applied to some molecules but not to others. “Metabolites” have no meaning here. Next, “Metabolites” “influenced by external factors” is again, a misuse of words. Biosynthesis and catabolism of polar molecules and lipids can be influenced by external factors but not the metabolites themselves.

Page 10, line 418. Figure 2. Abbreviation lists TGFbeta; however, there is no TGFbeta in the figure itself.

Provided Figure 2 must be redrawn. Metabolic pathway should be first leading to endotypes and ending in phenotypes (ECM deposition, etc). Major players characteristic to each endotype should be depicted (i.e. cytokines characteristic to Th2-high and Th2-low and leading to each phenotype, if possible).

Author Response

Dear Reviewer

Thank you kindly for taking the time to provide very useful comments regarding our submitted manuscript titled: “Application of metabolomics in obesity-related childhood asthma subtyping: a narrative scoping review”.

We have addressed all the points you kindly made and we have amended the text to reflect these changes and communicate the main message with greater clarity. Please see the answers to your comments below.

Comment 1

page 7 line 280. Table 1 (that should be Table 2) is missing

Answer to comment 1

Thank you for highlighting this.

This is Table 1 as we deleted the textbox.

We had attached this at the platform but it looks like this did not come together at the manuscript. We will make sure this appears on the manuscript this time.

Comment 2

Section C (page 7, line 284) – either to move somewhere at the beginning of the manuscript or eliminate as it brings little information to understand how metabolomics can help to identify asthma endotypes.

Answer to comment 2

Thank you very much for this suggestion. We eliminated all part C apart from the following sentences added at the Discussion section at the lines 377 to 386. See these sentences copied below:

With an optimistic tone, there are novel measurement platforms that can possibly address the above challenges in the future.  Semiconductor metal oxide-based chemo resistant sensors are being used for a variety of different functions (e.g., monitoring of the type and concentration of products of gas combustion) due to their small size, portability, and cost effectiveness [101]. These tools detect mediators (e.g., volatile organic compounds), specifically using exhaled breath. Agilent Seahorse is another novel measurement tool assessing the kinetic activity (i.e., rates) of two main ATP-producing pathways in live cells; the mitochondrial respiration and glycolysis. Consequently, the tool offers a dynamic rather than static measurement of metabolism and is currently used in live cell cultures and in tissue (i.e., bronchial tissue) samples [102].”

Comment 2

p.8 line 351. The use of the phrase “metabolites are sensitive molecules” is misleading. There is oxidation, enzymatic cleavage, hydrolysis, as a few examples of processes that can be applied to some molecules but not to others. “Metabolites” have no meaning here. Next, “Metabolites” “influenced by external factors” is again, a misuse of words. Biosynthesis and catabolism of polar molecules and lipids can be influenced by external factors but not the metabolites themselves.

Answer to comment 2

Thank you very much for this clarification. The text has been amended (manuscript _ track changes on, pages 393-403). Please see the amended text copied below:

The third challenge is associated with biospecimen type, collection and preservation.. “Detecting and measuring polar molecules and lipids with increased sensitivity, specificity and reproducibility is difficult. Factors that can impact detection (e.g., timing of biospecimen collection, biospecimen collection procedure, processing, stabilization, and storage) are also associated with sampling techniques. We might achieve consistent and more sensitive measurements (i.e., better detection yield) and easy validation of metabolomics-related biomarkers by utilizing novel sampling techniques (e.g., nasal scrapes, nasosorption). In summary, identifying metabolites as biomarkers for obesity-related childhood asthma remains an analytical challenge.”

Comment 3

Page 10, line 418. Figure 2. Abbreviation lists TGFbeta; however, there is no TGFbeta in the figure itself.

Response to comment 3

We apologize for this error. This has been amended now.

Comment 4

Provided Figure 2 must be redrawn. Metabolic pathway should be first leading to endotypes and ending in phenotypes (ECM deposition, etc). Major players characteristic to each endotype should be depicted (i.e., cytokines characteristic to Th2-high and Th2-low and leading to each phenotype, if possible).

Answer to comment 4

Thank you for this useful comment. We have redrawn this figure following your useful advice and we have inserted this inside the manuscript.

Round 2

Reviewer 2 Report

The article is still not prepared properly.

The structure of the article is still not prepared like a typical and widely accepted review article. Therefore, in my opinion, the article cannot be accepted for publication.

Author Response

Dear Reviewer

Thank you kindly for taking the time to provide comments regarding our submitted manuscript titled: “Application of metabolomics in obesity-related childhood asthma subtyping: a narrative scoping review”.

We fully respect your decision around rejection.

We also appreciate that we have been given the chance to respond to the comment(s) that are likely linked to this decision.

Comment 1

“The article is still not prepared properly.”

Answer to Comment 1

Thank you for describing your personal view. We understand you believe that the article is not prepared properly. It is however unclear to us which are the reasons. It reads as the following sentence describes the reason. Therefore, we address the following comment (i.e., Comment 2).

Comment 2

The structure of the article is still not prepared like a typical and widely accepted review article.

Answer to Comment 2

Thank you for this comment. In general, the synthesis and presentation of narrative reviews, in contrast to the systematic reviews, varies.

The structure of our narrative review is as follows:

  1. Introduction B. Relevant sections and subsections C. Discussion with main focus around possible challenges in the application of metabolomics in precision medicine in obesity-related childhood asthma D. Conclusion E. Future directions of research

We also cite the Journal’s guidelines around the structure of a narrative reviews:

MDPI | Article Types

“The structure can include an Abstract, Keywords, Introduction, Relevant Sections, Discussion, Conclusions, and Future Directions, with a suggested minimum word count of 4000 words”.

Although we believe that the manuscript’s structure is in line with the journal’s guidelines,  we will follow the editorial advice and decision and will make adjustments if this is required.

Reviewer 3 Report

Please, correct a few minor points.

Abstract. Too many “However”, “On the positive note”, “Nevertheless”. Please, simplify your narrative, be more categorical.

Line 253. There is no “secretion of long-chain n-3 fatty acids”, it is a decreased level of n-3 polyunsaturated fatty acids. Secretion means release from phosphor/neutral lipids due to the activity of phospholipases or lipases. Please, either specify or change.

Line 255-256. “Intervention with long chain fatty acids”… which one? n-3 or n-6? In the context of inflammation, you need to specify which ones were used here.

Author Response

Dear Reviewer

Thank you again kindly for your very useful comments aiming to improve our submitted manuscript.

We have addressed all the points you kindly made and we have amended the text to reflect these changes. Kindly find the answers to your comments below:

Comment 1

“Abstract. Too many “However”, “On the positive note”, “Nevertheless”. Please, simplify your narrative, be more categorical.”

Answer to Comment 1

Thank you for your comment. We have revised the abstract (lines 20-30) and have edited the text. Kindly see the edited text below:

“The heterogeneity of obesity-related asthma phenotypes may be attributable to the underlying pathogenetic mechanisms. Although few childhood obesity-related asthma endotypes have already been described, our knowledge in this field is incomplete. An evolving analytical profiling technique, metabolomics, has the potential to link individuals’ genetic backgrounds and environmental exposures (e.g., diet) to disease endotypes. This will ultimately help define clinically relevant obesity-related childhood asthma subtypes that respond better to targeted treatment. However, there are challenges related to this approach.”

Comment 2

“Line 253. There is no “secretion of long-chain n-3 fatty acids”, it is a decreased level of n-3 polyunsaturated fatty acids. Secretion means release from phosphor/neutral lipids due to the activity of phospholipases or lipases. Please, either specify or change.”

Answer to Comment 2

Thank you for this clarification. We have edited the text below:

Till now, the evidence shows that decreased peripheral blood n-3 long-chain fatty acid levels are associated with increased risk for obesity-related asthma”.

Comment 3

Line 255-256. “Intervention with long chain fatty acids”… which one? n-3 or n-6? In the context of inflammation, you need to specify which ones were used here.

Thank you for pointing this out. We have clarified as “n-6 long chain fatty acids”.
